# Photodynamic Inhibition of Herpes Simplex Virus 1 Infection by Tricationic Amphiphilic Porphyrin with a Long Alkyl Chain

**DOI:** 10.3390/pharmaceutics15030956

**Published:** 2023-03-15

**Authors:** Igor Jurak, Maja Cokarić Brdovčak, Lara Djaković, Ivana Bertović, Klaudia Knežević, Martin Lončarić, Antonija Jurak Begonja, Nela Malatesti

**Affiliations:** 1Department of Biotechnology, University of Rijeka, Radmile Matejčić 2, HR-51000 Rijeka, Croatia; 2Photonics and Quantum Optics Unit, Center of Excellence for Advanced Materials and Sensing Devices, Ruđer Bošković Institute, Bijenička Cesta 54, HR-10000 Zagreb, Croatia

**Keywords:** cationic amphiphilic porphyrin, HSV-1, photodynamic therapy, PACT

## Abstract

Photodynamic therapy (PDT) is broadly used to treat different tumors, and it is a rapidly developing approach to inactivating or inhibiting the replication of fungi, bacteria, and viruses. Herpes simplex virus 1 (HSV-1) is an important human pathogen and a frequently used model to study the effects of PDT on enveloped viruses. Although many photosensitizers (PSs) have been tested for their antiviral properties, analyses are usually limited to assessing the reduction in viral yield, and thus the molecular mechanisms of photodynamic inactivation (PDI) remain poorly understood. In this study, we investigated the antiviral properties of TMPyP3-C_17_H_35_, a tricationic amphiphilic porphyrin-based PS with a long alkyl chain. We show that light-activated TMPyP3-C_17_H_35_ can efficiently block virus replication at certain nM concentrations without exerting obvious cytotoxicity. Moreover, we show that the levels of viral proteins (immediate-early, early, and late genes) were greatly reduced in cells treated with subtoxic concentrations of TMPyP3-C_17_H_35_, resulting in markedly decreased viral replication. Interestingly, we observed a strong inhibitory effect of TMPyP3-C_17_H_35_ on the virus yield only when cells were treated before or shortly after infection. In addition to the antiviral activity of the internalized compound, we show that the compound dramatically reduces the infectivity of free virus in the supernatant. Overall, our results demonstrate that activated TMPyP3-C_17_H_35_ effectively inhibits HSV-1 replication and that it can be further developed as a potential novel treatment and used as a model to study photodynamic antimicrobial chemotherapy.

## 1. Introduction

Photodynamic therapy (PDT) is a treatment that uses a nontoxic photosensitizer (PS) that can be activated by a specific wavelength of light, usually from a laser or light-emitting diode (LED), to generate reactive oxygen species (ROS) and destroy cancerous cells or inactivate pathogens. Photosensitizers (PSs) in photodynamic antimicrobial chemotherapy (PACT) are rapidly developing and have been applied to inactivate and inhibit the replication of protozoa, fungi, bacteria, and viruses [1,2,3,4,5,6,7,8]. Viruses are the most diverse biological entities, and light-activated compounds that have been shown to be effective against some viruses may be ineffective against others. The exact molecular mechanism of viral inactivation by PACT is highly dependent on the PS applied, but these mechanisms are generally rather poorly investigated and largely limited to cytotoxic studies and viral yield reduction. Several studies have shown that enveloped viruses are more susceptible to PACT than nonenveloped viruses, indicating that the viral envelope is the primary target for PSs. Other mechanisms include direct damage to the genomic material and proteins of the virus, limiting the infectivity of the virus and its capacity to replicate [2,9,10,11,12]. On the other hand, the application of a PS can lead to compromised integrity of the infected cell, immune stimulation, and/or induction of a non-permissive state for viral replication [13,14]. Currently, little is known about antiviral mechanisms triggered by nontoxic levels of PS compounds [14]. Herpes simplex virus 1 (HSV-1) is an important human pathogen and is frequently used as a model in preclinical and clinical studies of antiviral PACT [2,15,16]. HSV-1 can cause a variety of different diseases, from clinically minor and self-limiting cold sores to life-threatening encephalitis, and it is a common cause of corneal disease and the leading infectious cause of ocular morbidity and blindness [17,18]. The currently available treatments against productive HSV-1 infections largely rely on nucleoside/nucleotide analogs such as acyclovir or its derivatives and viral DNA polymerase inhibitors (i.e., foscarnet) [19]. However, the use of target-based compounds ultimately leads to the emergence of resistant mutants, which significantly hinders future successful treatments, especially in immunocompromised patients and hematopoietic stem cell recipients [20,21,22,23,24]. Thus, there is an urgent need for the development of novel drugs and alternative approaches to treat the disease, and in particular, approaches in which the virus has limited possibilities to develop resistance, such as PACT. Indeed, the benefits of photodynamic therapy have already been demonstrated for the treatment of herpes labialis [25] and herpetic keratitis [26]. We have recently shown that TMPyP3-C_17_H_35_ (Figure 1), a tricationic porphyrin-based PS with a long alkyl chain, synthesized in our laboratory [27], effectively inhibits *Legionella pneumophila*, an environmental bacterium and an opportunistic pathogen that causes Legionnaires’ disease [28,29]. Compared to other tested PSs, the compound showed superior properties in blocking *L. pneumophila* replication and biofilm formation, indicating a potential application in disinfection [29]. The main aim of this study was to determine the potential of TMPyP3-C_17_H_35_ to inhibit the replication of HSV-1, which could promote the development of new PACT-based therapies against viral infections. We used TMPyP3-C_17_H_35_ and HSV-1 to investigate the mechanisms by which tricationic amphiphilic porphyrin-based compounds with long alkyl chains inhibit viral infection. We found that the compound was very effective in reducing the infectivity of free virus and inhibiting its replication at nM concentrations.

## 2. Materials and Methods

### 2.1. Cell Culture and Viruses

Vero cells (African green monkey kidney cell line) were obtained from the American Type Culture Collection (ATCC). The cells were cultured in Dulbecco’s modified Eagle medium (DMEM, Pan Biotech, Aidenbach, Germany) supplemented with 5% (*vol*/*vol*) fetal bovine serum (FBS, Pan Biotech, Aidenbach, Germany), 10^5^ U of penicillin/L, 0.1 g of streptomycin/L (Lonza, Walkersville, MD, USA), and 0.1 g/L of sodium pyruvate (Lonza, Walkersville, MD, USA) at 37 °C and 5% CO_2_. HSV-1 strain KOS was a generous gift from Donald M. Coen (Harvard Medical School). Virus stocks were prepared and titrated as previously described [30].

### 2.2. Photosensitizer and Light Source

The photosensitizer used in these experiments is 5-(4-octadecanamidophenyl)-10,15,20-tri(*N*-methylpyridinium-3-yl)porphyrin trichloride, TMPyP3-C_17_H_35_, which was previously prepared, characterized, and described by Malatesti et al. [27]. TMPyP3-C_17_H_35_ was dissolved in phosphate-buffered saline (PBS) and stored at 4 °C, protected from light, as a 2.4 mM stock solution. The light source was designed and calibrated at CEMS-Photonics and Quantum Optics Unit at the Ruđer Bošković Institute and based on light-emitting diodes (LEDs) and diffusers to provide homogenous irradiation at the surface of a standard cell culture plate, with a central emission wavelength of 643 nm (red light) [27]. The light fluence rate to which the cells were exposed was 2 mW cm^−2^. The light dose delivered to the cells in a 96-well plate for an exposure period of 15 min was 1.8 J cm^−2^.

### 2.3. Cytotoxicity of TMPyP3-C_17_H_35_

To test the cytotoxicity of TMPyP3-C_17_H_35_, Vero cells were seeded one day before the experiment at a density of 8 × 10^3^ cells per well of a 96-well plate. After 24 h of cultivation, the cells were treated with TMPyP3-C_17_H_35_ at different concentrations (4.8 nM to 4.8 µM) and incubated for 30 min. Subsequently, the medium was replaced with fresh growth medium, and the cells were mock exposed or exposed to light for 15 min. After the treatment, the cells were kept in the dark at 37 °C and 5% CO_2_ for the indicated time and metabolic activity was analyzed using a standard MTT assay as an indicator of the compound cytotoxicity assessed. Briefly, the growth medium was removed from the wells and 100 μL of tetrazolium dye MTT (3-(4,5-dimethylthiazol-2-yl)-2,5-diphenyltetrazolium bromide) (Carl Roth, Karlsruhe, Germany) prepared in growth medium at a final concentration of 0.5 mg/mL was added. The cells were then incubated for 4 h followed by the addition of 100 μL of dimethyl sulfoxide (DMSO) directly to the wells. The light absorbance was measured at 570 nm using a Sunrise microplate reader (Tecan, Mannedorf, Switzerland). Working dilutions of TMPyP3-C_17_H_35_ were always freshly prepared in growth medium on the day of the experiment. For the proliferation assay, 25,000 cells per well were seeded into a 48-well plate the day before treatment with the compound. The cells were washed and mock treated or treated with the indicated concentration of the compound for 30 min. The medium was then replaced with fresh growth medium, and the cells were exposed to light for 15 min or mock treated. At the indicated time points, the cells were trypsinized and stained with trypan-blue (Lonza), and viable cells were counted using a hemocytometer.

### 2.4. Western Blot Analysis

The cells for protein analysis were lysed in radioimmunoprecipitation assay buffer (RIPA; 50 mMTris-HCl (pH 8.0), 150 mM NaCl, 1 mM EDTA, 1% (*vol*/*vol*) Nonidet P-40, 0.5% sodium deoxycholate) with the addition of a complete protease inhibitor cocktail (Roche) on ice for 10 min. The cell lysates were precleared by centrifugation at 13,000× *g* for 10 min, and the proteins were resuspended in 2× Laemmli buffer (Biorad, Hercules, CA, USA) and heat-denatured at 95 °C for 6 min. The proteins were separated in 10% sodium dodecyl sulphate-polyacrylamide gels (SDS-PAGE), transferred to nitrocellulose membranes (Santa Cruz Biotechnology, Santa Cruz, CA, USA), and detected using mouse monoclonal antibodies: anti-actin (Chemicon International, Temecula, CA, USA), anti-ICP0 (Abcam, Cambridge, UK), and gC (Abcam). The proteins of interest were visualized using a horseradish-peroxidase-conjugated antibody (Cell Signaling, Danvers, MA, USA) and SuperSignal West Femto Maximum Sensitivity Substrate (Thermo Fisher Scientific, Waltham, MA, USA), and documented using the ChemiDoc XRS+ System (Bio-Rad, Hercules, CA, USA).

### 2.5. Photodynamic Inactivation of HSV-1

Inhibition of virus replication. Vero cells seeded in 12-well or 24-well plates one day before the experiment were mock treated or treated with TMPyP3-C_17_H_35_ at different concentrations (9.6 nM to 4.8 µM) and incubated for 30 min, after which the compound containing medium was replaced with fresh medium and the cells were mock infected or infected with HSV-1 at the indicated multiplicity of infection (MOI). After 30 min, the infected cells were exposed to light or kept in the dark for 15 min. One hour after infection, the infectious media were replaced with fresh growth medium, and the cells were further incubated at 37 °C and 5% CO_2_. At the indicated time points, the cells were fixed in 5% MetOH and 10% HAc solution and stained with Giemsa (5% Giemsa, Sigma, Singapore), and the number of plaque was determined, or supernatants were collected, and the viral titer was determined using a standard plaque assay [30]. 

Direct effect of TMPyP3-C_17_H_35_ on virus particle. A dilution of a virus stock in growth medium (~1 × 10^7^ Pfu) was mock treated or treated with TMPyP3-C_17_H_35_ at final concentrations of 4.8 μM, 1.2 μM, 480 nm, 240 nM, 48 nM, or 9.6 nM and exposed to light for 15 min or kept in the dark. The infectious virus in treated samples was assayed by a standard plaque assay [30].

### 2.6. Microscopy

To test the cellular localization of TMPyP3-C_17_H_35_, Vero cells were seeded on coverslips one day before the experiment at a density of 5 × 10^4^ cells per well of a 24-well plate. After 24 h of cultivation, the cells were treated with 21 µM of TMPyP3-C_17_H_35_ and incubated for the indicated periods of time. The coverslips were then briefly rinsed with PBS and prepared for microscopy. Images were obtained with a Zeiss LSM880 confocal laser scanning microscope (Carl Zeiss, Oberkochen, Germany) using a 63× Plan Apochromat object.

## 3. Results

### 3.1. Toxicity of TMPyP3-C_17_H_35_ in Vero Cells

Cationic amphiphilic porphyrins, such as TMPyP3-C_17_H_35_ (Figure 1), rapidly enter cells and can be localized in different subcellular compartments where they exert cytotoxic activity (reviewed in [1]). To monitor the cellular uptake of TMPyP3-C_17_H_35_, we treated Vero cells, cells commonly used in HSV-1 assays, with 21 μM solution of TMPyP3-C_17_H_35_ for up to 120 min and analyzed them by confocal microscopy. Notably, TMPyP3-C_17_H_35_ absorbs light in the broad UV-Vis spectrum and emits light in the red part of the spectrum (maximum absorption and emission are at 422 and 650 nm, respectively) [27]. 

After only 30 min, we observed an evenly distributed punctate staining around the cells, which appeared as aggregates on the cell membrane (Figure 2). Longer incubation with the compound resulted in punctate staining throughout the cell but distant from the nucleus, whereas prolonged incubation resulted in the accumulation of aggregates proximally to the nucleus (usually more intense at one site), with diffuse staining throughout the cytoplasm (Figure 2). These structures resemble membrane vesicles and could represent lysosomes, the Golgi apparatus, and/or the endoplasmic reticulum or other structures in the cytoplasm, which is consistent with other studies on similar compounds [31,32]. It is important to note that the treatment of Vero cells with such high concentrations of TMPyP3-C_17_H_35_ for an extended period of time resulted in increased vacuolization and cell death. We are very cautious in interpreting these results because we were unable to directly visualize the compound at lower concentrations, which would represent a more physiologically relevant transport of the compound; nevertheless, our results show that the compound can be efficiently internalized (Figure 2). 

Next, we aimed to investigate the antiviral properties of TMPyP3-C_17_H_35_, particularly concentrations at which the compound does not cause significant cellular toxicity. To investigate the cytotoxicity of TMPyP3-C_17_H_35_, we treated Vero cells with a wide range of concentrations of the compound (4.8 nM to 4.8 μM; Figure 2) and analyzed the metabolic activity of the treated cells using a standard MTT assay. Briefly, cells were seeded in 96-well plates and mock treated or treated with TMPyP3-C_17_H_35_ for 30 min. Following the treatment, the growth medium or medium containing TMPyP3-C_17_H_35_ was replaced with fresh medium to remove the excess of the compound and incubated for an additional 30 min to allow internalization of the compound and to mimic the conditions in the infectious experiments (details below). Subsequently, one set of the plates was irradiated with a low dose (2 mW cm^−2^) of red light (643 nm) for 15 min, while the other set was kept in the dark. The cytotoxicity of the compound was assessed 24 h after light exposure. Cells treated with the lower concentrations of TMPyP3-C_17_H_35_ (4.8 nM–480 nM) showed no apparent signs of cytotoxicity of the compound regardless of activation by light (Figure 3). In addition, this result also shows that the application of a low dose of red light for 15 min causes no obvious harm to the cells. On the other hand, cells treated with increasing concentrations of TMPyP3-C_17_H_35_ (1.2 μM–4.8 μM) and irradiated showed a striking loss of metabolic activity (from 90% to less than 25% after irradiation), compared with cells also treated but not irradiated (sustained metabolic activity between 80 and 90%) (Figure 3A), indicating dose- and light-dependent cytotoxicity. We observed a very similar pattern of cytotoxicity in human neuroblastoma cell lines SH-SY5Y, HeLa [27], and HCT116 [33]. TMPyP3-C_17_H_35_ at concentrations greater than 10 μM was generally toxic to cells regardless of irradiation. These results are comparable to other porphyrin-based PSs tested by others [34,35]. However, in our study, we were primarily interested in investigating the antiviral properties of the PS compound at subtoxic concentrations; assays such as the MTT assay are quite robust and cytotoxicity may not be as evident. Therefore, we also examined the effect of TMPyP3-C_17_H_35_ on Vero cell proliferation at lower concentrations (9.6–240 nM) for 72 h. We observed no obvious changes in cell morphology and only a limited effect on cell proliferation at higher concentrations (Figure 3B), and therefore chose these concentrations as our main focus.

### 3.2. TMPyP3-C_17_H_35_ Inhibits Replication of HSV-1

To initially test the antiviral properties of TMPyP3-C_17_H_35_, we treated Vero cells with a broad range of concentrations of the compound (9.6 nm–4.8 μM) for 30 min (i.e., to allow internalization of the compound). The treated cells were briefly washed and infected with a very small amount (<100 Pfu) of wild-type HSV-1 strain KOS. The infection was left to proceed for 30 min (i.e., to allow virus entry), followed by a 15 min light activation of the PS. Then, the infectious medium was removed and the cells were overlaid with a viscous solution of methylcellulose to prevent the formation of secondary plaques. Plaque formation was analyzed 72 h after infection (h p. i.) (Figure 4A). 

As expected, in the mock treated cells, viral infection resulted in dozens of plaques, whereas in the cells treated with TMPyP3-C_17_H_35_ and irradiated, we observed a dose-dependent decrease in the number of plaques (Figure 4B). At the highest concentration (4.8 μM and 1.2 μM), light-activated TMPyP3-C_17_H_35_ inhibited viral replication below the sensitivity of the assay. At this concentration, TMPyP3-C_17_H_35_ had no significant effect on viral replication in the absence of light activation (Figure 4B), which was expected because we did not observe significant cytotoxicity of the compound at these concentrations without light activation (Figure 3). Importantly, in addition to the reduced number of plaques, we observed an obvious reduction in plaque size in cells treated with 480 nM TMPyP3-C_17_H_35_ and irradiated, compared with untreated or treated but not irradiated cells at any concentration (Figure 4C). This phenomenon was not observed at a TMPyP3-C_17_H_35_ concentration of 240, 48, or 9.6 nM. These results indicate that the activation of an internalized PS in the early stages of infection strongly inhibits the virus from initiating replication and/or might create non-permissive conditions for HSV-1 replication. 

To further investigate the observed antiviral activity, we performed infections in the presence of TMPyP3-C_17_H_35_ at concentrations between 9.6 nM and 1.2 μM, with or without irradiation, and measured the virus yield. Briefly, Vero cells were pretreated with TMPyP3-C_17_H_35_ for 30 min, then the medium containing the compound was replaced with fresh medium, and cells were infected either as mock infection or with HSV-1 at an MOI of 1, and 30 min after being irradiated for 15 min. One hour after infection, virus inoculum was replaced with fresh medium and the cells were incubated for an additional 18 h. The supernatants were then collected, and the number of infectious virions in the supernatant was determined by titration on the Vero cells. Similarly to the previous experiment, where we did not observe any significant effect of the non-activated compound on plaque development, the virus yield was also only slightly affected. On the other hand, the application of activated TMPyP3-C_17_H_35_ at concentrations of 1.2 μM, 480 nM, and 240 nM reduced the number of infectious units in a dose-dependent manner, i.e., from below the detection limit (>100,000×) to 100× and 10×, respectively, compared with untreated cells (Figure 5).

The compound at a concentration of 48 nM and 9.6 nM showed no effect on the virus yield in both conditions. Taken together, these results indicate a strong efficacy of light-activated TMPyP3-C_17_H_35_ in inhibiting virus infection at certain nM concentrations. 

### 3.3. TMPyP3-C_17_H_35_ Reduces Levels of of Immediate Early Proteins 

Different compounds can inhibit viral replication through a variety of different mechanisms, but little is known about the exact mechanisms of how PSs inhibit viral infection. For example, it is possible that the observed reduced viral yield is due to the virus being unable to initiate its gene expression because of damage to the virion (DNA or capsid), an inability to replicate its DNA, or other changes in the infected cell. To first clarify at which phase of productive infection TMPyP3-C_17_H_35_ inhibits HSV-1 replication (before or after DNA replication), we analyzed the levels of the immediate early protein ICP0 (an important viral protein whose expression does not depend on viral DNA replication) and gC (a late viral protein that depends on viral DNA replication) during the course of infection. Similarly to the previous experiment, Vero cells were pretreated with TMPyP3-C_17_H_35_ at various concentrations (1.2 μM, 480 nM, 240 nM, 48 nM, and 9.6 nM) and infected and irradiated, and samples for Western blot were collected at the indicated time points. Our results show that both viral proteins were strongly reduced in the cells treated with the activated compound at a concentration of 1.2 μM (Figure 6). We also observed some level of dark activity of TMPyP3-C_17_H_35_ on ICP0 and gC expression, compared with the untreated cells (Figure 6). These results are consistent with the results obtained using plaque and replication assays (Figure 4 and Figure 5). Somewhat surprisingly, at lower concentrations (480 nM and 240 nM) of TMPyP3-C_17_H_35_, we observed only a slight difference between activated and non-activated PSs regarding levels of both proteins, which cannot explain a more dramatic decrease in viral yield in the supernatant of the equally treated samples (10–100×) (Figure 5). Both the viral genes analyzed are strongly expressed during productive HSV-1 infection and thus Western blot analysis may not provide sufficient resolution. One can speculate that the compound triggers processes that could lead to inefficient viral maturation or activate antiviral defense mechanisms, which could explain this discrepancy. Nonetheless, our results clearly indicate that the photoactivated PS, which is internalized before the onset of infection, causes an early block in viral infection, probably before DNA replication. 

### 3.4. Addition of TMPyP3-C_17_H_35_ Early in Infection Inhibits HSV-1 Replication

In our experiments, we showed that the compound internalized prior to infection can block virus replication. However, there are many possible underlying mechanisms that could contribute to the observed early block in viral replication, including both entry and post-entry events. To initially investigate whether TMPyP3-C_17_H_35_ can effectively block viral replication after virus internalization, we infected cells with an MOI of 1 and allowed the infection to proceed for at least 30 min. This time period is sufficient for most virions to internalize and even initiate viral gene expression. After that, cells were treated at different times after infection (from 30 min to 3 h p. i.) with 240 nM or 9.6 nM TMPyP3-C_17_H_35_ for 30 min, washed, and then irradiated. Virus yields were measured 18 h p. i. We chose a concentration of 240 nM of the compound because at this concentration we observed no toxicity but some degree of antiviral activity (Figure 3B and Figure 4), and 9.6 nM served as a negative control. As expected, and in agreement with other results, the concentration of 9.6 nM had no effect on viral replication regardless of light activation. Surprisingly, the light-activated compound at a concentration of 240 nM successfully inhibited viral replication (reduction >100×) only when the cells were treated shortly after internalization, i.e., by adding the compound immediately after removal of the infectious medium (30 min p. i.) up to 2 h p. i. The inhibitory effect of the compound decreased with increasing time post infection, and by 3 h p. i., the effect on virus yield was minor or absent (Figure 7).

In addition, we observed no inhibitory effect on viral replication when cells were treated and exposed to light prior to infection (240 nM), indicating that the compound at nontoxic concentrations does not induce antiviral mechanisms that may limit infection (Figure 7). 

Overall, our results show that TMPyP3-C_17_H_35_ can efficiently suppress viral replication when applied after the onset of infection; however, the application window is rather limited. Our results also indicate that early events, i.e., before virus replication, are affected by activated TMPyP3-C_17_H_35_. At this point, we cannot rule out the possibility of host targets for the observed inactivation. It has been observed previously that PSs can damage endocytic vesicles during virus entry [36]; however, the role of endocytosis in herpesvirus infection and its contribution in Vero cell infection, compared to membrane fusion, has not been thoroughly investigated [37,38,39]. Another possibility is that activated TMPyP3-C_17_H_35_ leads to damage of the viral capsid and/or viral DNA, preventing efficient gene expression and replication of the virus.

### 3.5. TMPyP3-C_17_H_35_ Decreases HSV-1 Infectivity

We demonstrated that non-activated TMPyP3-C_17_H_35_ did not significantly decrease cell viability (Figure 3) even at high concentrations (4.8 μM) (Figure 3). On the other hand, the toxicity of the activated compound is pronounced only at concentrations of 1.2 μM or higher. However, antiviral properties can also be observed at lower concentrations (Figure 4, Figure 5, Figure 6 and Figure 7). To investigate whether TMPyP3-C_17_H_35_ can directly affect the infectivity of free virions, we prepared approximately 1 × 10^6^ Pfu in 1 mL of growth medium per sample and treated with different concentrations of TMPyP3-C_17_H_35_ with or without irradiation. The treated infectious medium was used to determine the number of remaining infectious virions using a standard dilution plaque assay (Figure 8). 

Interestingly, we observed a dose-dependent reduction (i.e., >10^5^×–10×) in the number of plaque-forming units in the suspension treated with activated TMPyP3-C_17_H_35_ at 240 nM or higher. The treatment of virions with 48 nM and 9.6 nM activated compound had no effect on infectivity, nor did any non-activated compound at any concentration, event at the highest concentration (4.8 μM). This result was not surprising, since it has been previously reported that PSs can damage viral envelopes, capsid proteins, and nucleic acids (reviewed in [1,2]). Nonetheless, our results show that intercellular inhibitory effects on virus replication and direct effects on virus infectivity can be obtained in a similar range of concentrations (i.e., above 240 nM). Additionally, we did not observe smaller plaques or replication defects when replicative viruses were further passaged after treatment. These results indicate that TMPyP3-C_17_H_35_ impairs HSV-1 virions, probably by damaging the virion membrane and limiting its infectivity, and that DNA can be excluded as the main target of TMPyP3-C_17_H_35_ at the tested concentrations. 

## 4. Discussion

Recent outbreaks of diverse viral infections, including coronaviruses (SARS-CoV-1, -2, MERS-CoV-1), filoviruses (EBOLA virus), flaviruses (HCV, ZIKA, etc.), and hepatitis viruses (HCV, HVA, HVB, etc.) have greatly accelerated research on antivirals, particularly in the area of repurposing and broad-spectrum antivirals. The development of highly efficient photosensitizers and light sources (lasers and LED-based) underscored the potential of antimicrobial photodynamic chemotherapy (PACT) in the treatment of viral infections and the photodynamic disinfection of blood products (reviewed in [2]). In particular, the development of chemically defined and non-mutagenic porphyrin-based cationic amphiphilic compounds with limited dark toxicity and the ability to penetrate membranes overcomes most of the limitations previously associated with PACT. Antiviral PDTs have been applied in numerous clinical studies of recurrent HSV-1 infections, and, although this has been largely successful and promising, standardized protocols have yet to be established [25,40,41]. In this study, we investigated the potential of water-soluble TMPyP3-C17H35, a porphyrin-based tricationic PS that has shown excellent ^1^O_2_ production and PDT efficacy in suppressing *L. pneumophila* in our previous study [28,29], to inhibit HSV-1 replication. We observed relatively rapid internalization (<30 min) of TMPyP3-C17H35 into Vero cells and its accumulation into structures that resemble lysosomes. Although the excited compound emits red fluorescence, we could not directly visualize the compound at concentrations below 20 μM, which significantly limited the analysis. The compound tested aggregates at the high concentrations used for imaging, so the observed internalization and subsequent subcellular localization should be interpreted with caution, as they may not be relevant, owing to different uptake mechanisms between aggregates and soluble compounds (e.g., pinocytosis vs. active endocytosis). Porphyrin-based cationic compounds have been found localized mainly in lysosomes, endoplasmic reticulum, Golgi apparatus, or mitochondria, which is highly important for understanding their antiviral properties [42,43], but the exact molecular mechanisms of virus inactivation under low cytotoxic conditions remain to be discovered. Tetracationic *meso*-tetra (*N*-methyl-4-pyridyl)porphyrin (TMPyP4) was shown, for instance, to localize in lysosomes but also to change localization after irradiation, which often happens in PDT and which, moreover, can enhance its effect [44]. Nonetheless, determining the precise subcellular localization is beyond the scope of this study. Another important limitation was that the experiments were not performed in the dark, i.e., limited exposure of the compound to bright light could not be entirely avoided. Such limited exposure showed no obvious effects on cell viability, but some contribution to measured biological effects (i.e., viral replication) cannot be excluded. 

Although we found that the compound induces massive cell death when activated by light at concentrations >1 μM, low systemic toxicity was observed in the nanomolar range, which is comparable to other similar compounds [45,46,47]. This allowed us to design antiviral assays at subtoxic concentrations, which is highly important to avoid toxicity concerns and to extend the potential application to the inactivation of different viruses in sensitive biological materials (e.g., serum or blood). However, in a number of experiments, we used high concentrations of the compound to demonstrate its potential for broader application, including cancer treatment, and to serve as a positive control for virus inactivation. It is important to note that the potential genotoxicity and mutagenicity of porphyrin-based compounds are considered low [48], and there are no reports of PDT causing secondary tumors, but any such concerns can be circumvented with localized application of the compound and light [49]. 

In our study, we were unable to determine the precise molecular targets of HSV-1 inactivation by photoactivated TMPyP3-C_17_H_35_, and several lines of evidence indicate that there are multiple and distinct mechanisms. First, using a simple plaque assay in which we monitored individual plaques that developed from the spread of virus to neighboring cells from a single infected cell, we observed a dose-dependent decrease in the number of plaques in cells treated with the compound before infection and irradiated after infection. In addition, we observed an obvious reduction in the size of the plaques that developed. These results are very puzzling and may indicate that the internalized and activated compound (a) inactivates or reduces the ability of the virus to replicate and (b) has prolonged effect on viral replication by inducing a less permissive state of the cell. Damage to the viral genome could explain the reduction in the number or size of plaques, but not the wt replication properties of these viruses when transferred to untreated cells (Figure 7), suggesting that the outer virion structures, such as envelopes or capsids [2], may be the main viral targets. Indeed, we show that the infectivity of free virions is significantly reduced by treatment with the compound (Figure 8), and we also observed that surviving viruses can replicate as the parental virus These results indicate that the viral genome was not significantly damaged. In addition, at concentrations that are not toxic to infected cells, TMPyP3-C_17_H_35_ can efficiently block virus infection when applied early post infection, strongly suggesting a viral target for the compound. Smetana et al. have observed a similar property of phthalocyanines to inhibit HSV-1 when applied up to 45 min post infection [10]. On the other hand, 5-aminolavulionic acid (5-ALA) showed strong antiviral properties in the late post-adsorption period (>3 h p. i.) [50]. It would be interesting to determine the target for this inactivation. However, it should also be noted that treatment of cells with TMPyP3-C_17_H_35_ at higher concentrations that induce general cytotoxicity inhibits viral replication regardless of when it is applied post infection. 

The observed reduction in plaque size (Figure 4) can also be explained by TMPyP3-C_17_H_35_ targeting cellular structures and creating unfavorable conditions for virus replication. Although we did not observe obvious signs of cell death, we cannot exclude possible damage to mitochondria or other structures [51] that led to a less permissive state of the cells in this assay. The inability of the compound to inhibit viral replication when added at a later time point in infection may be explained by its effect on an unknown critical component that has no function late in infection, for example, virus factors involved in trafficking. Overall, we show complex mechanisms behind the TMPyP3-C_17_H_35_ inactivation of HSV-1 probably involving viral and host targets. Our results also indicate that the mechanism behind the inactivation might be dramatically different at different concentrations. Nonetheless, further work is needed to elucidate the main targets of cationic porphyrin-based photosensitizers to better optimize their antiviral properties. 

Clearly, the clinical application of PDT in the treatment of viral diseases faces many challenges, yet its use has been proposed for various viruses [52]. However, despite a vast number of compounds having been tested in preclinical studies, clinical trials are still rather rare, so the information on the benefits of such treatments is limited. The treatment of patients with frequently occurring cold sores and viral ocular diseases (e.g., herpetic keratitis) with local application of the drug and light source has already been shown to provide significant benefits to patients [15,25,26,53]. Advances in this area are particularly important for patients treated with first-line antiviral drugs, such as acyclovir (ACV), who do not respond to treatment because of viral resistance. Moreover, the tested compound, TMPyP3-C_17_H_35_, may be a candidate for the treatment of warts caused by human papillomavirus (HPV), for which PDT has provided significant benefits to patients [54]. However, concerted efforts in standardizing protocols and selecting promising candidates are needed to achieve these goals.

## 5. Conclusions

Based on this in vitro study, we conclude that TMPyP3-C_17_H_35_, a tricationic amphiphilic porphyrin-based photosensitizer (PS) with a long alkyl chain, could be utilized to inhibit HSV-1 infection. Our results demonstrate the potential of the compound tested to inhibit free virus and its replication in infected cells at concentrations that do not provoke significant toxicity (<1 μM). Our results indicated that different mechanisms might contribute to HSV-1 inactivation under these conditions, namely, the direct inactivation of virions, the inhibition of virus replication early in infection, and effects on the permissiveness of infected cells. The compound also effectively inhibits HSV-1 replication at concentrations exceeding 1 μM, likely due solely to the induction of cell death. Further studies are needed to reveal the exact mechanisms of viral inhibition, and testing in in vivo models is required to determine the clinical potential of the compound.

## Figures and Tables

**Figure 1 pharmaceutics-15-00956-f001:**
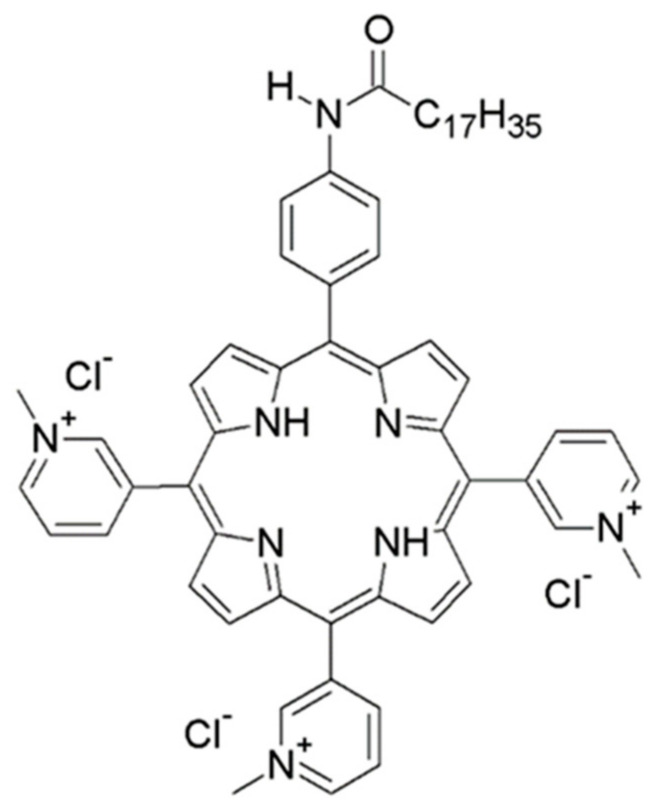
Chemical structure of TMPyP3-C_17_H_35._

**Figure 2 pharmaceutics-15-00956-f002:**
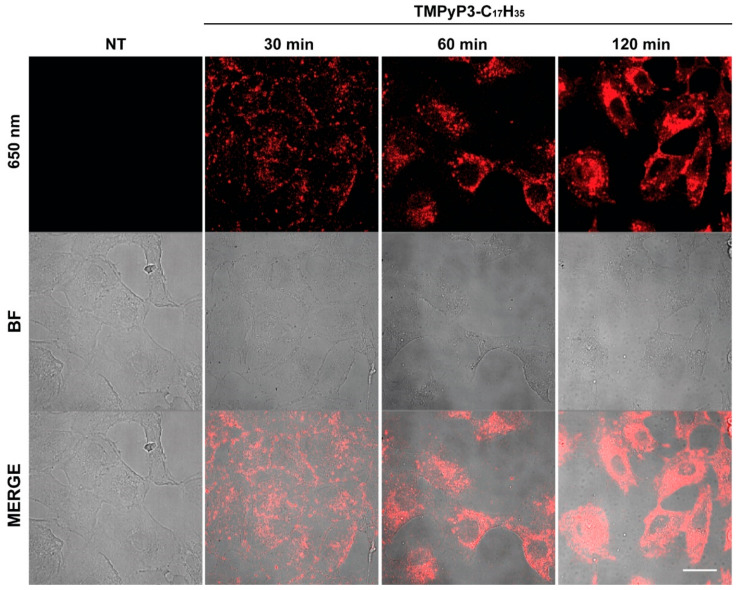
Internalization of TMPyP3-C_17_H_35_ in Vero cells. Vero cells were mock treated or treated with 21 µM TMPyP3-C_17_H_35_ for indicated periods of time and analyzed by confocal laser scanning microscopy. Scale bar, 20 µm. Upper panel emission at 650 nM. BF-bright filed.

**Figure 3 pharmaceutics-15-00956-f003:**
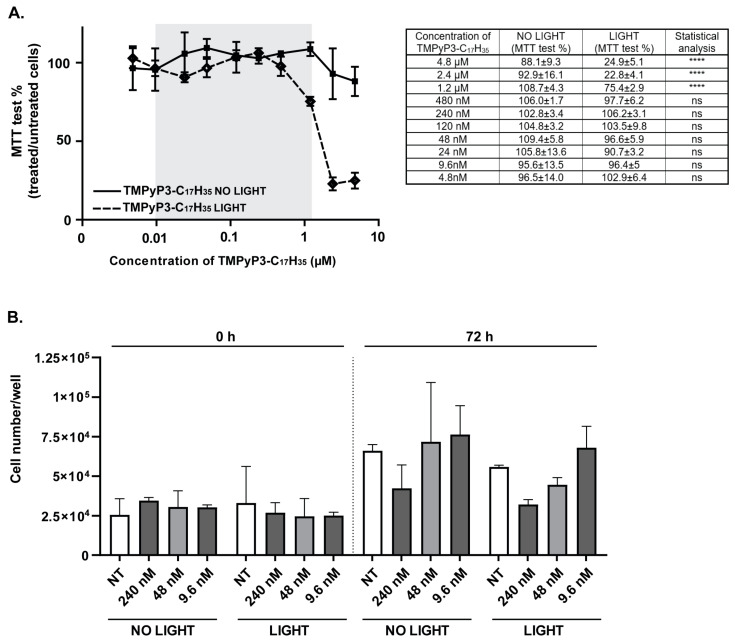
Cytotoxicity of TMPyP3-C_17_H_35_. (**A**) Vero cells were treated with a wide range of concentrations of TMPyP3-C_17_H_35_ (4.8 nM–4.8 µM). One set of the plates was irradiated with low-fluency red light (643 nm, 2 mW/cm^2^) (LIGHT), and the other set was kept in the dark (NO LIGHT). Metabolic activity was determined after 24 h using MTT assay. The table (right panel) shows the values shown in the graph (left panel). All samples were tested in quadruplicates. Data were analyzed using the Mann–Whitney U test; *p* values indicate significant differences (**** *p* < 0.0001, not significant ns *p* >0.05). Shaded area represents range of concentrations used in the experiments that include infection. (**B**) Vero cells were mock treated or treated with TMPyP3-C_17_H_35_ (9.6 nM–240 nM). One set of the plates was irradiated with low-fluency red light (643 nm, 2 mW/cm^2^) (LIGHT), and the other set was kept in the dark (NO LIGHT). Cell numbers per well were determined after 72 h. All samples were tested in triplicate.

**Figure 4 pharmaceutics-15-00956-f004:**
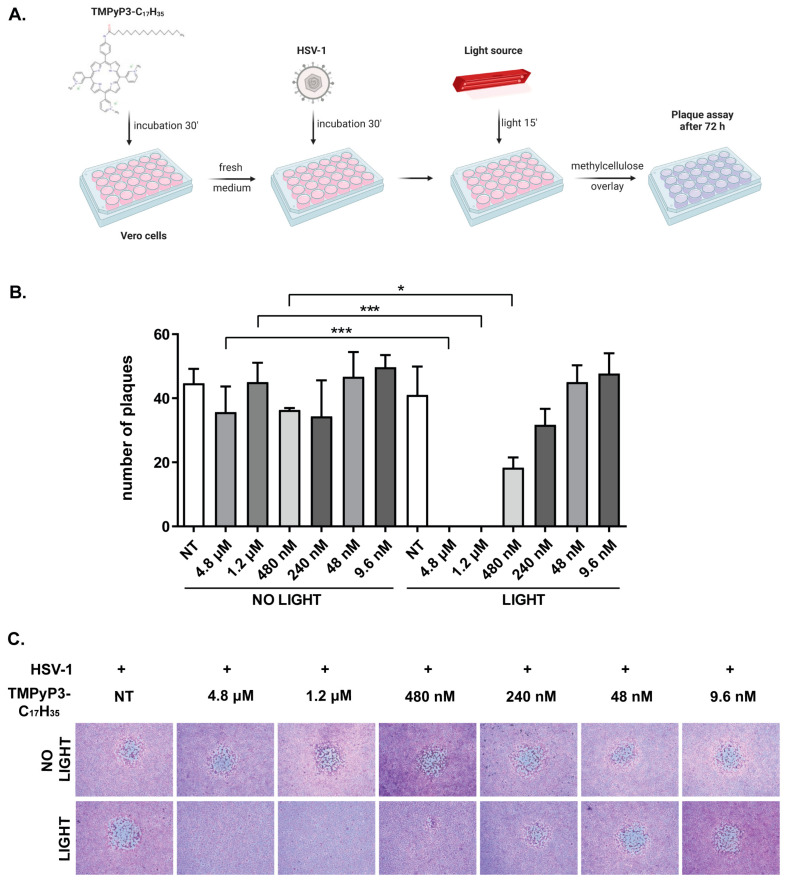
TMPyP3-C_17_H_35_ inhibits replication of HSV-1. Vero cells were treated with a wide range of concentrations of TMPyP3-C_17_H_35_ (9.6 nM–4.8 µM), infected with HSV-1 at a very low MOI and irradiated. One hour after infection, cells were overlaid with methylcellulose and left for an additional 72 h to develop plaques; cells were then fixed and stained, and the number of plaques was determined. (**A**) Schematic representation of the experiment. (**B**) The number of plaques developed after treatment with different TMPyP3-C_17_H_35_ concentrations. (**C**) Images of the fixed plaques were obtained using inverted microscope at 50× magnification. All samples were tested in triplicate and data were analyzed using ANOVA, post-hoc Turkey’s multiple comparison test (*p* values indicate significant differences; *** *p* = 0.0001, * *p* < 0.05).

**Figure 5 pharmaceutics-15-00956-f005:**
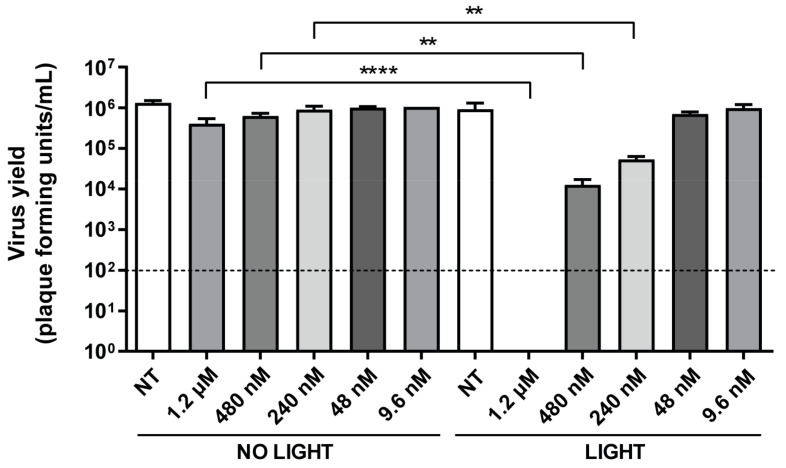
Light-activated TMPyP3-C_17_H_35_ inhibits HSV-1 infection at nM concentrations. Vero cells were infected after treatment with TMPyP3-C_17_H_35_ at concentrations between 9.6 nM and 1.2 µM, with or without irradiation. After 18 h of infection, supernatants were collected and the number of plaque-forming units (Pfu) determined by titration. NT: not treated. All samples were tested in triplicate, and data were analyzed using unpaired t-test (*p* values indicate significant differences; **** *p* = 0.0001, ** *p* = 0.0048). Dashed line in the graph represents detection limit of the experiment of 100 Pfu/mL.

**Figure 6 pharmaceutics-15-00956-f006:**
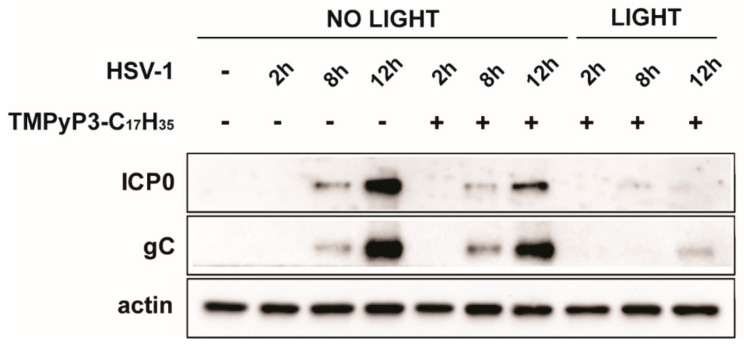
TMPyP3-C_17_H_35_ reduces levels of immediate early and late proteins. Vero cells were pretreated with 1.2 µM concentration of TMPyP3-C_17_H_35_, infected with MOI 1, and TMPyP3-C_17_H_35_ activated or not activated with light. Samples were collected at 2 h, 8 h, and 12 h post infection and probed by immunoblotting with antibodies against viral immediate–early protein ICPO and late protein gC.

**Figure 7 pharmaceutics-15-00956-f007:**
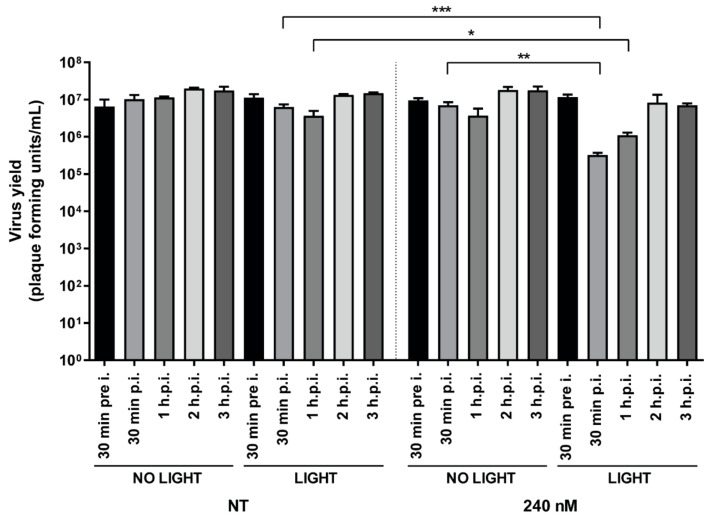
Addition of TMPyP3-C_17_H_35_ early in the infection inhibits HSV-1 replication. Vero cells pretreated (30 min prior to infection) with the compound, irradiated and infected, or infected and treated at indicated hours post infection (p. i.). At 18 h after infection, supernatants were collected and the number of plaque-forming units determined by titration. NT: not treated. All samples were tested in triplicate and data were analyzed using unpaired t-test (*p* values indicate significant differences; *** *p* = 0.0006, ** *p* = 0.0015, * *p* = 0.025).

**Figure 8 pharmaceutics-15-00956-f008:**
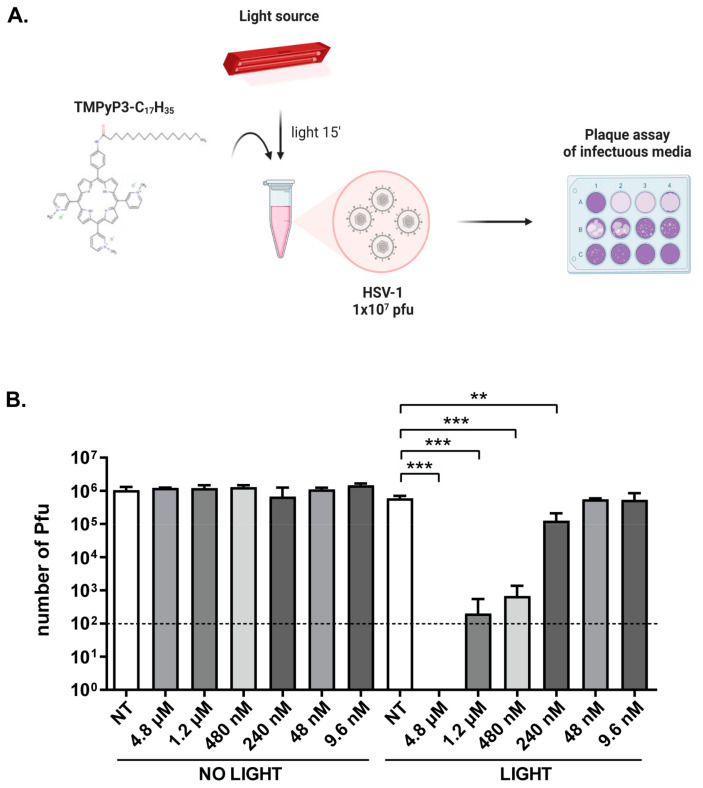
Direct effect of TMPyP3-C_17_H_35_ on HSV-1 virions. A suspension of HSV-1 (~1 × 10^6^ virions) was treated with TMPyP3-C_17_H_35_ at final concentrations of 4.8 μM, 1.2 μM, 480 nM, 240 nM, 48 nM, or 9.6 nM, and exposed to a light source. (**A**) Schematic representation of the experiment. (**B**) The number of plaque-forming units in sample was determined by plaque assay. All samples were tested in triplicate, and data were analyzed using unpaired t-test (*p* values indicate significant differences; *** *p* = 0.0008, ** *p* = 0.0046). The dashed line in the graph represents the detection limit of the experiment (100 Pfu/mL).

## Data Availability

The data presented in this study are available on request from the corresponding author.

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
