# Peer review of "Photodynamic Inhibition of Herpes Simplex Virus 1 Infection by Tricationic Amphiphilic Porphyrin with a Long Alkyl Chain"

_pharmaceutics, 2023, doi:10.3390/pharmaceutics15030956_

Round 1
Reviewer 1 Report
The introduction is too long, please try to concentrate on the main elements.
The aim of the study is not clearly stated
The ethical approval and the Committee approval number should be stated for studies on cells.
Figure 1 is of high quality.
It describes the Internalization of TMPyP3-C17H35 in Vero cells.
Figure 2 describes the Cytotoxicity of TMPyP3-C17H35 in light versus nonlight. I suggest modifying the x-axis to be more readable. Also for figure 3B, 4, 7B.
Figure 3C is of high quality, Images of the fixed 260 plaques were taken using inverted microscope at 10x magnification. Scale bar, 200 μm.
The discussion is comprehensive. I suggest adding limitations of the study.
The conclusion should be clearly stated.
Author Response
We thank you for your critical reading of the manuscript. We greatly appreciate your efforts and suggestions. We have improved our manuscript based on your suggestions and comments. Below you will find, point by point, our responses to all your comments.
Thank you
- The introduction is too long, please try to concentrate on the main elements.
Thank you for this comment. We have shortened and modified the introduction accordingly (track change).
- The aim of the study is not clearly stated.
We appreciate this comment. We have added the aims of the study (the last paragraph of the Introduction).
- The ethical approval and the Committee approval number should be stated for studies on cells.
The presented study was performed in the compliance with current state regulation, and the Ethical committee approval was not required. The Vero cell line used in the experiment was derived from African green monkey kidney cells and are standardly used in HSV-1 assays.
- Figure 1 is of high quality. It describes the Internalization of TMPyP3-C17H35 in Vero cells.
Thank you for this comment.
- Figure 2 describes the Cytotoxicity of TMPyP3-C17H35 in light versus nonlight. I suggest modifying the x-axis to be more readable. Also for figure 3B, 4, 7B
Thank you for this comment. We have added an additional table (Figure 2A) with concentrations to make the results more readable.
- Figure 3C is of high quality, Images of the fixed 260 plaques were taken using inverted microscope at 10x magnification. Scale bar, 200 μm.
Thank you for this comment.
- The discussion is comprehensive. I suggest adding limitations of the study. The conclusion should be clearly stated.
Thank you for these comments. We have included the limitations of the study (Lines 406, 420) and introduced a new section 5. Conclusions.
Reviewer 2 Report
The work is focused on the study of the antiviral properties of TMPyP3-C17H35, a tricationic amphiphilic porphyrin-based PS, against HSV-1 replication, when irradiated at nanomolar concentration. The authors reported the photodynamic effect induced by TMPyP3-C17H35 in the inhibition of the virus replication and tried to report and evaluate the cellular uptake.
Unfortunately, the work is not easily to read, the presented data are confusing, and it is necessary to move up and down in the paper to try to understand the results. In addition, there are some references to previous works, but no specific data or a brief summary are reported, so the comparison is not pleasant and comprehensible.
I suggest authors to deeply review the work and extensively modify the results and discussion sections before accepting for publication in Pharmaceutics.
Some comments:
- Extensive revision of the English and form, the manuscript is full of errors and typos (i.e affiliation 2 after Croatia in the second affiliation, line 123, line 158, line 198, line 211, line 223, line 235, line 245, line 250 and so on)
- Most of the nM concetrations became nm, which is something completely different.
- In the 3.1 results section: line 198, line 207 and line 211. The concentrations are not clear, the authors have to indicate the right concentration and not a range, because it is really difficult to understand. I can also suggest adding some tables, to better itemize the real used concentration.
- Figure 2, can the authors reproduce the graphs by normalizing each value on the cell number? Probably it could be easier to understand.
- In section 3.1 the authors concluded with: “Therefore, we decided to use the subtoxic concentrations of TMPyP3-C17H35 (i.e. between 9.6 nM – 1.2 uM) (Fig. 2; shaded area) in our further experiments” and in the section 3.2 the authors started with “To first test the antiviral properties of TMPyP3-C17H35, we treated Vero cells with a 234 broad range of concentrations of the compound (6 nm – 4 uM)”. Considering that at concentrations higher than 1.2 uM the compound is toxic, why the authors decided to investigate the higher concentration? Please, clarify this point.
- Line 271, the authors declared: “Consistent with the previous result, TMPyP3-C17H35 270 at a concentration of 1.2 uM reduced the number of infectious units below the detection limit”. Can the authors specify which previous results? I’m not able to find any other experiments on the replication inhibition with the 1.2 uM.
- In line 263 the authors reported: “we performed infections in the presence of TMPyP3-C17H35 at concentrations between 10 nm and 1.2 μM”, but later in the text they mentioned 0.096 uM (line 274, and figure 4), which correspond to 100 nM. Which concentration did the authors tested, 10 or 100nM?
- In section 3.3, can the authors reported the data related to all the other concentration (maybe in the SI)? This could help to better understand. Moreover, in Figure 5, the compound seems to be toxic at 1.2 uM after 12h no light, which is the opposite compared to the experiment reported in figure 2A. Can the authors clarify this point?
- Section 3.4, why the authors decided to test 240 nM concentration among all the others?
- The discussion section is not so easy to read, can the authors try to resume some results or summarize the most relevant results with the reported concentrations?
- The conclusion section is missed. Can the authors add this section, summarizing the main results and the advancements compared to the literature?
- Error bars are missed in figure 3B and 7B, and significances are missed in all the presented data. The authors have to add them.
- The structure of the TMPyP3-C17H35 is not readable in both figure 3A and 7A. Increase the size and resolution.
Author Response
The work is focused on the study of the antiviral properties of TMPyP3-C17H35, a tricationic amphiphilic porphyrin-based PS, against HSV-1 replication, when irradiated at nanomolar concentration. The authors reported the photodynamic effect induced by TMPyP3-C17H35 in the inhibition of the virus replication and tried to report and evaluate the cellular uptake.
Unfortunately, the work is not easily to read, the presented data are confusing, and it is necessary to move up and down in the paper to try to understand the results. In addition, there are some references to previous works, but no specific data or a brief summary are reported, so the comparison is not pleasant and comprehensible.
I suggest authors to deeply review the work and extensively modify the results and discussion sections before accepting for publication in Pharmaceutics.
We thank you for your critical reading of the manuscript and the comprehensive analysis that has helped us to improve it. We have made a great effort to change the results and the discussion section to make them more readable. We have added a new section 5. Conclusions, which will help highlight the specific objectives and results.
The specific comments are listed below point-by-point.
Some comments:
- Extensive revision of the English and form, the manuscript is full of errors and typos (i.e affiliation 2 after Croatia in the second affiliation, line 123, line 158, line 198, line 211, line 223, line 235, line 245, line 250 and so on). Most of the nM concetrations became nm, which is something completely different.
We thank you for this comment. We apologize to the reviewer. We made some errors in formatting the manuscript into journal format that we did not notice. Nevertheless, we have thoroughly revised the manuscript to improve any language deficiencies, including concentration errors.
- In the 3.1 results section: line 198, line 207 and line 211. The concentrations are not clear, the authors have to indicate the right concentration and not a range, because it is really difficult to understand. I can also suggest adding some tables, to better itemize the real used concentration.
Thank you very much for this comment. We apologize to the reviewer for the sloppiness in formatting the symbol "micro" that we did not notice after the journal formatting. We have added a new table to Figure 2 to help understand the exact concentrations of the compounds used.
- Figure 2, can the authors reproduce the graphs by normalizing each value on the cell number? Probably it could be easier to understand.
We agree with the criticism that Figure 2 contains complex information, so we have rewritten the results section to make it more understandable. However, we have not changed the figure itself.
- In section 3.1 the authors concluded with: “Therefore, we decided to use the subtoxic concentrations of TMPyP3-C17H35 (i.e. between 9.6 nM – 1.2 uM) (Fig. 2; shaded area) in our further experiments” and in the section 3.2 the authors started with “To first test the antiviral properties of TMPyP3-C17H35, we treated Vero cells with a 234 broad range of concentrations of the compound (6 nm – 4 uM)”. Considering that at concentrations higher than 1.2 uM the compound is toxic, why the authors decided to investigate the higher concentration? Please, clarify this point.
Thank you for that point. We agree that the paragraphs lack coherence. We have modified the sections 3.1 and 3.2. Our main focus in this study was on subtoxic concentrations, but we also used high concentrations as a positive control for the inhibition in our experiments. At high concentrations, the antiviral property is largely based on induction of cell death, which is well documented in numerous studies, whereas at subtoxic concentrations the mechanisms are not clear and very intriguing.
- Line 271, the authors declared: “Consistent with the previous result, TMPyP3-C17H35 270 at a concentration of 1.2 uM reduced the number of infectious units below the detection limit”. Can the authors specify which previous results? I’m not able to find any other experiments on the replication inhibition with the 1.2 uM.
Thank you for this comment. We agree that the statement is confusing and unclear and there is an obvious error. The result refers to Figure 3 B, in which virus replication was not detected in the cells treated with 1 microM compound and irradiated (right); in contrast to not-irradiated but treated cells (left panel; a small number of plaques)). In the original figure the concentration was erroneously shown as 1 microM and has been corrected. The compound at 4 mM concentration prevented development of plaques regardless of light activation. The exact concentrations have been corrected throughout the manuscript, we apologize for the confusion.
- In line 263 the authors reported: “we performed infections in the presence of TMPyP3-C17H35 at concentrations between 10 nm and 1.2 μM”, but later in the text they mentioned 0.096 uM (line 274, and figure 4), which correspond to 100 nM. Which concentration did the authors tested, 10 or 100nM?
Thank you very much for this comment. We apologize for the confusion. There is an error in Figure 4. The exact concentration used in the experiments was 0,0096 mM (i.e. 9,6 nM). Concentrations have been corrected throughout the manuscript.
- In section 3.3, can the authors report the data related to all the other concentration (maybe in the SI)? This could help to better understand.
Thank you very much for this comment. We are willing to provide the full screening panel upon request. Nevertheless, we cannot explain this discrepancy, but we suspect that the Western blot does not provide sufficient resolution for the difference in selected genes at selected times post infection. Also, these genes are expressed quite strongly in infected cells which limits the resolution. Nonetheless, our results show that both classes of genes are reduced; immediate early (IPC0) and late (gC), indicating block that prevents expression of IE genes (certainly before replication of DNA).
- Moreover, in Figure 5, the compound seems to be toxic at 1.2 uM after 12h no light, which is the opposite compared to the experiment reported in figure 2A. Can the authors clarify this point?
Thank you for that question. Indeed, we did not observe toxicity with not-activated compound at 1,2microM using the MTT assay, however we did observe clear inhibition of viral replication under the same conditions. On the one hand, we believe that the observed “dark-activity” can be attributed to low dose of light during manipulation, which unfortunately could not be avoided in our experimental settings. On the other hand, the replication assay could be more sensitive and could also indicate antiviral activity of non-activated compound at this concentration. We have included this in the discussion Lines 316-330.
- Section 3.4, why the authors decided to test 240 nM concentration among all the others?
Thank you for this question. We agree that selection of the compound needs an explanation. We have modified the text line 347. We chose a concentration of 240 nM the compound because at this concentration we observed no toxicity but some degree of antiviral activity (Fig. 3B and 4).
- The discussion section is not so easy to read, can the authors try to resume some results or summarize the most relevant results with the reported concentrations? The conclusion section is missed. Can the authors add this section, summarizing the main results and the advancements compared to the literature?
Thank you for this comment. We have added a new “Conclusion” section.
- Error bars are missed in figure 3B and 7B, and significances are missed in all the presented data. The authors have to add them.
Thank you for this comment. Figure 3B is showing the representative experiment and was not performed in multiplicate. Figure 7B (average of titrations in duplicates) and 7C (infected in triplicates and each titrated in duplicate) shows representative coupled experiment. Infection of cells and subsequent titration shown in 7C was performed to increase sensitivity of 7B to detect residual low level of infectious units.
- The structure of the TMPyP3-C17H35 is not readable in both figure 3A and 7A. Increase the size and resolution.
Thank you for this note. We have corrected the figure accordingly and added a new panel in Figure 1. showing the structure of the compound.
Reviewer 3 Report
The manuscript presented for review describes very interesting results of research on the foto-destruction of herpes simplex virus 1.
I recommended that this manuscript be published in the presented form after introducing corrections regarding the concentration units of the tested photosensitizer (here TMPyP3C17H35).
Author Response
The manuscript presented for review describes very interesting results of research on the foto-destruction of herpes simplex virus 1.
I recommended that this manuscript be published in the presented form after introducing corrections regarding the concentration units of the tested photosensitizer (here TMPyP3C17H35).
Thank you for the critical reading of the manuscript. We apologize for the formatting issues with the symbol for the “micro”. Concentrations have been corrected throughout the manuscript.
Reviewer 4 Report
The authors reported on several mechanisms contributing to photodynamic inhibition of herpesvirus replication by tricationic amphiphilic porphyrin with a long alkyl chain. Their analyses were scientifically designed and examined. Their results indicated several mechanism might contribute to photodynamic inhibition of HSV-1 replication. Followings are my questions and comments.
L220: The authors described "at lower concentrations (10 - 250 nM) ." However, the concentrations were "0.096, 0.048 and 0.24 microM" in Figure 2B. Were the lower concentrations 9.6 (see my comment below) - 240 nM?
L229-230: The authors described that Vero cells were ... (9.6 nM - 240 nM). However, concentrations were 0.096 microM in B of Figure 2. They should be 0.0096 microM.
L234-236: The authors described that we treated .. of the compound (6 nM - 4 microM). The authors described that they decided to use the subtoxic concentrations of TMPyP3-C17H35 (i.e. between 9.6 nM -1.2 microM) in Lines 222 to 223. Why did the authors used greater concentrations of the reagent than subtoxic concentrations? And also the authors described 6 nM - 4 microM as a broad range of concentrations of the compound. However, the concentrations examined in Figure 3B were 4, 1, 0.25 and 0.06 microM, i.e. 60 nM. Which is correct, 6 nM or 60 nM?
L250-252: The authors described that they observed an obvious reduction in plaque size. If so, the authors should show quantitative data on the plaque sizes and statistical analysis.
L257-261: In Figure 3B, it seems that the number of plaques in no reagent with irradiation were greater than that without irradiation. What does this data mean?
L266:267: It seems that some words were deleted between the line 266 and 267. Please clarify it.
L 282-286: The authors should examine statistical analyses and show significance of differences among data.
L302: The authors described that these results were consistent with the results obtained in the titration assays. However, data on virus yields shown in Figure 4 indicated that no pfu at 1.2 microM with no irradiation. Therefore the results were no consistent between Figure 4 and Figure 5. Clarify and explain this discrepancy.
L405-413: The authors described that the compound induced massive cell death when activated by light at concentrations > 1 microM. If so, why did the authors examined the experiments at concentrations >1 microM such as Figure 3, Figure 4 and Figure 5. Clarify and explain about this inconsistency.
Author Response
The authors reported on several mechanisms contributing to photodynamic inhibition of herpesvirus replication by tricationic amphiphilic porphyrin with a long alkyl chain. Their analyses were scientifically designed and examined. Their results indicated several mechanism might contribute to photodynamic inhibition of HSV-1 replication. Followings are my questions and comments.
We thank you for your critical reading of our manuscript. We appreciate your efforts to improve the manuscript. We have addressed all the points raised and described them point by point below.
- L220: The authors described "at lower concentrations (10 - 250 nM) ." However, the concentrations were "0.096, 0.048 and 0.24 microM" in Figure 2B. Were the lower concentrations 9.6 (see my comment below) - 240 nM?
- L229-230: The authors described that Vero cells were ... (9.6 nM - 240 nM). However, concentrations were 0.096 microM in B of Figure 2. They should be 0.0096 microM.
Thank you for this comment. We apologize for sloppiness with the concentrations. The concentrations have been corrected throughout the manuscript.
- L234-236: The authors described that we treated of the compound (6 nM - 4 microM). The authors described that they decided to use the subtoxic concentrations of TMPyP3-C17H35 (i.e. between 9.6 nM -1.2 microM) in Lines 222 to 223. Why did the authors used greater concentrations of the reagent than subtoxic concentrations? And also the authors described 6 nM - 4 microM as a broad range of concentrations of the compound. However, the concentrations examined in Figure 3B were 4, 1, 0.25 and 0.06 microM, i.e. 60 nM. Which is correct, 6 nM or 60 nM?
Thank you for that point. We agree that the submitted manuscript lacked coherence. We have amended the results section and corrected numerous errors that did not indicate the exact concentrations. A number of errors went unnoticed after formatting the manuscript in journal form. Our main focus in this study was on subtoxic concentrations, but we also used high concentrations as a positive control for inhibition in our experiments. At high concentrations, the antiviral property is largely due to induction of cell death, which is well documented in numerous studies, whereas the mechanisms at subtoxic concentrations are not clear and, in pour opinion, are very intriguing.
- L250-252: The authors described that they observed an obvious reduction in plaque size. If so, the authors should show quantitative data on the plaque sizes and statistical analysis.
We thank you for this comment and suggestion. In addition to reduction in plaque number, we indeed have observed reduction in plaque size. At this point we cannot explain the molecular mechanism behind this phenomenon because individual plaques develop from a single infectious unit (i.e. plaque forming unit). We have not observed inherited deficiency of viruses produced from such small plaques (they replicated to wt levels if passaged further) indicating that genome was not affected and that this phenomenon is likely due to antiviral effect of the compound on the host cells (i.e. triggering antiviral state). We have not measured the size of plaques but just collected several representative photos shown the phenomenon. Lines 448-456.
- L257-261: In Figure 3B, it seems that the number of plaques in no reagent with irradiation were greater than that without irradiation. What does this data mean?
Thank you for this comment. Figure 3B shows one of the representative experiments in which very simple assays were used to initially determine the antiviral potential of the compound. We believe that the slightly higher number of plaques shown in Figure 3 in irradiated cells compared to non-irradiated cells is not a significant observation, but an experimental fluctuation. Indeed, our other similar experiments do not show this trend.
- L266:267: It seems that some words were deleted between the line 266 and 267. Please clarify it.
We have carefully rewritten a large part of the manuscript, and some changes have been made in this area as well. After formatting the manuscript into journal format by the editors, some formatting problems may have occurred; which hopefully are now resolved.
- L 282-286: The authors should examine statistical analyses and show significance of differences among data.
Thank you for this comment. We have included statistical analysis wherever was appropriate (Fig, 2, 4, 6, and 7). We have indicated in the figure legend whether the results show a representative experiment with not statistical power.
- L302: The authors described that these results were consistent with the results obtained in the titration assays. However, data on virus yields shown in Figure 4 indicated that no pfu at 1.2 microM with no irradiation. Therefore the results were no consistent between Figure 4 and Figure 5. Clarify and explain this discrepancy.
We thank you for this observation. As noted by the reviewer, we did not detect PfU after treatment with 1.2 microM of the unactivated compound (i.e., it was below the detection limit of 100 Pfu/ml), and we observed some gene expression in the Western blot assay. It is important to note that the detection limit was set high for two reasons: a) low sample volume; b) to avoid any effects of residual compound on the titration assay. However, we observed the same pattern at lower concentrations (i.e., 480 nM). Western blot analysis showed a much lower degree of inhibition than the titration assay. We think that these two assays resolve the phenomenon of inhibition differently, but also that we cannot exclude the possibility that processes such as egress may be affected by the compound, which would not be seen in our WB assays. It would be very interesting to explore these possibilities, but this is beyond the scope of the current study. We have modified section 3.3 to make it more readable.
- L405-413: The authors described that the compound induced massive cell death when activated by light at concentrations > 1 microM. If so, why did the authors examined the experiments at concentrations >1 microM such as Figure 3, Figure 4 and Figure 5. Clarify and explain about this inconsistency.
Thank you for pointing this out. Indeed, we emphasize the antiviral properties of the compound under subtoxic concentration and we use high, toxic, concentrations. Clearly, the compound has a strong antiviral activity in microM range, which was somewhat expected form such compounds, and we confirmed this. These results indicate a potential of the compound for further preclinical studies. However, understanding of the mechanisms that block viral replication, that are not based on destruction of cells, are largely unknown and, from the research point of view, neglected. High concentrations of the compound also served as a positive control in our experiments. We have modified the Discussion and introduced the Conclusion section to explain our aims, results and hopefully clarify inconsistencies.
Reviewer 5 Report
The submitted manuscript reports on tricationic amphiphilic porphyrin with a long alkyl chain for photodynamic inhibition of herpes simplex virus infection. This topic is of interest for readers of Pharmaceutics. However, I have some reservations about of the experimental design and data presented. I therefore recommend publication of this manuscript only if the authors can address the major issues noted below.
1. The title of this manuscript is “Several Mechanisms Contribute to Photodynamic Inhibition of Herpes Simplex Virus 1 Infection by Tricationic Amphiphilic Porphyrin with a Long Alkyl Chain”. However the mechanism studies are a very small component of this manuscript and are not systematically studied. The results are mainly based on the observation of cell viability and viral replication. So the title is not appropriate, while the significance is poorly addressed.
2. The introduction section needs to be improved. It needs to be clearly stated what are key research gaps, which are not achieved in the literature.
3. Error bars are missing in several figures. Also it is important to carry out statistical analysis. This information is not provided for all the results.
4. Have the authors repeated the western blot data and done any quantitive analysis (densitometry) for the results? Better to provide these results.
5. Additional mechanism studies are needed to demonstrate the effects of photodynamic therapy.
6. Animal studies should be carried out to demonstrate the effects on the infected tissues.
Author Response
The submitted manuscript reports on tricationic amphiphilic porphyrin with a long alkyl chain for photodynamic inhibition of herpes simplex virus infection. This topic is of interest for readers of Pharmaceutics. However, I have some reservations about of the experimental design and data presented. I therefore recommend publication of this manuscript only if the authors can address the major issues noted below.
- The title of this manuscript is “Several Mechanisms Contribute to Photodynamic Inhibition of Herpes Simplex Virus 1 Infection by Tricationic Amphiphilic Porphyrin with a Long Alkyl Chain”. However the mechanism studies are a very small component of this manuscript and are not systematically studied. The results are mainly based on the observation of cell viability and viral replication. So the title is not appropriate, while the significance is poorly addressed.
Thank you for your critical review of the manuscript. We agree with the reviewer that we have not addressed the exact mechanism by which the compound tested inhibits viral replication, but we show that the compound can inhibit viral replication by multiple mechanisms and at concentrations that induce limited toxicity. These include: Induction of cell death (higher concentrations), direct reduction of infectivity (free virions at nM concentrations); suppression of replication (after cellular uptake; at nM concentrations; in the context of limited cell toxicity); effect on cellular permissiveness (at nM concentrations; in the absence of cell toxicity). We did not identify specific targets of activated PS. The main message of the work is that PS can inhibit viral replication without inducing massive cell death, which opens additional possibilities for the use of PS in the treatment of viral infections. We have rewritten the manuscript to make it more readable and to emphasize the conclusions and significance2.
- The introduction section needs to be improved. It needs to be clearly stated what are key research gaps, which are not achieved in the literature.
Thank you for this note, we make significant changes in introduction putting more emphasis on, what is in our opinion, the key research gaps.
- Error bars are missing in several figures. Also it is important to carry out statistical analysis. This information is not provided for all the results.
Thank you for this note, we have included error bars in all experiments in which was appropriate Fig. 2, 4, 6, and 7 (some experiments show representative experiment which is also indicated Fig 3, Fig. 7B)
- Have the authors repeated the western blot data and done any quantitive analysis (densitometry) for the results? Better to provide these results.
- Additional mechanism studies are needed to demonstrate the effects of photodynamic therapy.
Thank you for comment. We strongly agree with the reviewer that additional research is needed to identify the exact molecular targets, however such investigation is beyond the scope of this study and requires significant additional resources.
- Animal studies should be carried out to demonstrate the effects on the infected tissues.
We agree with the reviewer that an animal study would greatly enhance the significance of our study, but at this time we cannot meet this criterion. Nevertheless, we believe that our limited study will stimulate interest in the field and allow further research.
Reviewer 6 Report
The article is devoted to the study of the photochemical activity of alkyl-substituted phthalocyanine in relation to HIV-1. The authors performed a wide range of studies from the cytotoxicity of phthalocyanine to their photochemical activity at various concentrations of the photosensor. From my point of view, the article is well written. Research methods are described in sufficient detail. The results are presented clearly. I recommend the article for publication.
Minor remarks:
Throughout the text, mkL and µM should be corrected
Some errors in lines 455-457
Author Response
The article is devoted to the study of the photochemical activity of alkyl-substituted phthalocyanine in relation to HIV-1. The authors performed a wide range of studies from the cytotoxicity of phthalocyanine to their photochemical activity at various concentrations of the photosensor. From my point of view, the article is well written. Research methods are described in sufficient detail. The results are presented clearly. I recommend the article for publication.
Minor remarks:
Throughout the text, mkL and µM should be corrected
Some errors in lines 455-457
Dear Reviewer,
Thank you for your critical review of our manuscript. We greatly appreciate your effort. We apologize for errors in concentrations. We have corrected these and other typos throughout the manuscript. We have also used an extensive language correction service to improve the quality of the manuscript.
Round 2
Reviewer 2 Report
The authors corrected and revised the manuscript according to the comments. However, some lack are still present such as the single representative experiment on cell assays.
Author Response
The authors corrected and revised the manuscript according to the comments. However, some lack are still present such as the single representative experiment on cell assays.
Thank you for your comments and suggestions for improving our manuscript. We appreciate your opinion very much. We also agree with the reviewer that the manuscript has its weaknesses and strengths. These include the presentation of the individual representative experiments. We performed the same experiments several times and with different cell lines. The results led to the same conclusions. Nevertheless, we are aware of the limitations of such experiments, and therefore the description of the results is more descriptive. At this stage, a significant investment of time and resources is required to repeat these experiments at the desired level. We are aware of the language flows and imperfections, we thank you for this comment, so the revised manuscript has been edited by the MDPI language service.
Reviewer 4 Report
The authors corrected and revised the manuscript appropriately.
Author Response
The authors corrected and revised the manuscript appropriately.
Dear reviewer,
Thank you for carefully reading of the manuscript and suggestions how to improve the manuscript.
Reviewer 5 Report
The authors have addressed my comments. Happy for it to be published.
Author Response
The authors have addressed my comments. Happy for it to be published.
Dear Reviewer,
Thank you for your critical reading and your suggestions and comments, which have helped us to improve the manuscript. We are aware of the weaknesses and strengths of the manuscript, including the quality of the English language. We will improve the manuscript with the help of a professional editor.
Thank you